# Tourism Development and Italian Economic Growth: The Weight of the Regional Economies

Giorgio Colacchio [ID] and Anna Serena Vergori *

Department of Law, Economics Division, University of Salento, 73100 Lecce, Italy
* Correspondence: serena.vergori@unisalento.it

**Abstract:** This research aims to study the relationship between economic growth and the increase in the tourism sector in Italy. Unlike most of the literature, we use the value added in the main economic sectors involved in tourism activity as a proxy for tourism development. The use of the tourism value added allows us to analyze the effect of both international and domestic tourism on per capita GDP growth. The main working hypothesis we tested is whether the relationship between GDP growth and the expansion of the tourism sector is in any way influenced by the geographic area referenced and/or the time period considered. Accordingly, we conducted our analysis at both the national and subnational (cluster) levels, splitting the original sample into two equal subperiods (1997–2008 and 2009–2019). The panel VAR analysis shows that for the country as a whole, tourism growth depends on the past value of the economic growth rate, especially for the subperiod 2009–2019. The cluster analysis clarifies that these outcomes are strongly determined by the cluster that covers the wealthiest Italian regions.

**Keywords:** tourism sector; economic growth; EDTG hypothesis; panel VAR models

## 1. Introduction

Tourism is not identifiable in a single unitary industry; indeed, its economic effects are distributed among many sectors, such as accommodation, food services, recreational services, and transportation, to cite a few typical tourism industries (Colacchio and Vergori 2022). The difficulty in defining a clear border for tourism activities raises some problems in measuring tourism's economic impact. On the one hand, it is rather difficult to accurately measure the share of output produced by tourism industries that is consumed by tourists. On the other hand, tourists also buy goods and services provided by industries that are not classified as tourism. Even the Tourism Satellite Account (TSA), developed by the UNWTO to standardize the measurement of the phenomenon among countries, is not comprehensive (Jenkins 2022).

The difficulties in measuring the economic impact of tourism mentioned above have not prevented scholars from studying the effects of tourism growth on economic development. In the last two decades, indeed, since the seminal contribution by Balaguer and Cantavella-Jordà (2002), the relationship between international tourism and economic growth and its causal direction have been widely investigated. The tourism-led economic growth hypothesis is at the core of this strand of literature that, in turn, has its theoretical background in the literature related to the export-led growth hypothesis. International tourism is a non-standard kind of export, characterized by the fact that it is the consumer that moves, not the product. Proving the tourism-led economic growth (TLG) hypothesis would mean that tourism could be one of the main determinants of overall long-run economic growth. In this case, a unidirectional causality from tourism to economic growth would exist. In addition to the TLG hypothesis, theoretical and empirical contributions have highlighted another possible causality direction going from economic growth to tourism. The economy-driven tourism growth (EDTG) hypothesis, in fact, suggests that

economic growth, and more generally economic development, causes tourism to expand. According to this view, the increase in physical and human capital related to economic growth can lead, for example, to the improvement of tourism-related infrastructure and service quality, while at the same time generating positive spillover effects on the tourism industry from other economic sectors (Lin et al. 2019).

Next to the TLG and EDTG hypotheses, empirical studies sometimes prove the validity of two other hypotheses: the feedback hypothesis, which supports a bidirectional causality between tourism and economic growth, and the neutrality hypothesis, according to which no relationship between the two variables exists.

Along this research line, the focus is usually on international tourism (using foreign tourism expenditures as a proxy), thus neglecting the crucial role that domestic tourism plays in some economies. In doing so, the main risk is underestimating tourism's contribution to economic growth. We aim to fill this gap by proposing to use tourist GDP as a proxy for tourism development. In this way, we achieve two goals. First, we consider the whole tourism phenomenon; second, we use an economic variable to estimate tourism development over time. In this respect, indeed, it is worth noting that in the literature, non-economic variables (such as the number of tourists visiting a destination) have been generally used when the aim was to estimate the economic impact of both domestic and inbound tourism (e.g., Cortés-Jiménez 2008; Colacchio and Vergori 2022).

Our research also contributes to the literature through the investigation of a possible different impact of tourism on territories characterized by varying levels of economic development. Although this issue has been raised (Lin et al. 2019), it needs to be studied in more depth. Accordingly, we propose a panel VAR analysis for Italian data at the regional level (NUTS 2) for the period 1997–2019. The data at hand allow us to estimate the relationship between tourism and economic growth for some regional clusters, showing that the outcomes obtained for the nation are strongly determined by the cluster that covers the wealthiest Italian regions. Furthermore, we study the impact of economic crises on the relationship under investigation by splitting the entire sample into two subperiods: 1997–2008 and 2009–2019—the last one, as is well known, was characterized by economic crises.

In the following section, we review the main literature that analyzes the relationship between tourism and growth in Italy. Then, after briefly describing the Italian regions' economies, we perform several pre-estimation tests, among which is the Granger causality test. Subsequently, we study panel VAR (PVAR) models for Italy, taking the entire sampled period and two subperiods into account along with the regional clusters obtained through the K-means algorithm. Finally, we discuss the results and offer conclusions.

## 2. Literature Review

The relevance of the topic and the attention paid by many scholars have led to the publication of various literature reviews about the relationship between tourism and economic growth (Ivanov and Webster 2013; Pablo-Romero and Molina 2013; Brida et al. 2016; Fonseca and Sanchez-Rivero 2019; Comerio and Strozzi 2019). Although such a relationship has generally been confirmed, the results of the causality nexus do not show complete agreement among scholars, sometimes even for the same country. As discussed in the literature (e.g., Antonakakis et al. 2015; Shahbaz et al. 2018), the Granger causality linkages between tourism development and economic growth are not stable over time, both in terms of magnitude and direction, mainly because of structural changes in the time series. Accordingly, even studies concerning the same country could get different results because of different methodologies employed and/or of different data considered.

In the following, without claiming to be exhaustive, we focus on some studies having Italy (also) as a case study. The real GDP per capita is generally used to measure economic growth. International tourist arrivals, international tourism receipts, and tourism expenditure are the three variables commonly used as proxies of tourism growth, singly, separately, or combined in weighted indexes.

Proença and Soukiazis (2008) use the Barro and Sala-i-Martin conditional convergence approach to confirm the tourism-led growth hypothesis for four southern European countries, claiming that tourism is a factor of convergence for the standard of living of those countries. They use panel data for the period 1990–2004, and the proxy for tourism is international tourism receipts data. An aggregate production function, which expresses GDP growth as a function of physical capital, human capital, and international tourism receipts, is at the core of the model used by Cortés-Jiménez and Pulina (2010) to investigate the impact of inbound tourism on the Italian and Spanish economies. In this model, the Granger causality test confirms the tourism-led growth hypothesis for Italy for the period 1954–2000.

Massidda and Mattana (2012), unlike most of the papers, focus only on Italy using quarterly data from 1987 to 2009. Through the analysis of a Structural Vector Error Correction Model, they investigate the relationship between real GDP, international per capita tourism arrivals, and total international transactions as a share of GDP. In the short run, the relationship that emerges is unidirectional from GDP to tourist arrivals, whereas, in the long run, GDP growth and tourism expansion are characterized by a bidirectional relationship (feedback hypothesis).

A unidirectional causal nexus from economic growth to international tourism receipts is found by Aslan (2013) for Italy through a Granger causality approach for a panel of twelve Mediterranean countries from 1995 to 2010. Tugcu (2014) uses annual panel data from 1998 to 2011 for twenty-one countries that border the Mediterranean Sea, performing a panel Granger causality test applied to both international tourism receipts and expenditures as proxies of the value added generated by tourism. The results of Tugcu's analysis validate the TLG hypothesis for Italy: both tourism expenditures and receipts positively affect economic growth. Shahbaz et al. (2018) aim at investigating the time-varying causal nexus between economic growth and tourism for the world's top ten international tourism destinations from the first quarter of 1990 to the fourth quarter of 2015, through a bootstrap rolling window Granger causality approach. They use a weighted index of international tourist arrivals, receipts, and tourism expenditure as a tourism development indicator. Italy is one of the few countries that exhibits the most robust bidirectional causal relationship between growth and tourism, although its magnitude is stronger for the relationship between real GDP and tourism development. The same database used by Shahbaz et al. (2018) is also analyzed by Shahzad et al. (2017). This latter study applies the quantile-on-quantile methodology to demonstrate that the link between tourism and economic growth also depends on the economic phases of expansion or recession. For Italy and most of the countries analyzed, the main result of Shahzad et al. is that the link between tourism and economic growth is stronger during recession periods. The feedback hypothesis between tourism and economic growth has been proven by Dogru and Bulut (2018) for seven European countries that border the Mediterranean Basin; the period analyzed was from 1996 to 2014, and the variable used to represent tourism economic expansion was tourism receipt growth rates.

As far as we know, Cortés-Jiménez (2008) and Colacchio and Vergori (2022) are the few studies that deal with the matter at a regional level and consider the whole phenomenon, not only international tourism. The first one uses panel data for Italian and Spanish regions from 1990 to 2000; it considers the effect of international and domestic tourism, proxied respectively by the number of nights spent by non-residents and residents. The empirical results show that both international and domestic tourism are relevant for coastal regions' economic convergence. In contrast, domestic tourism alone is a crucial factor for internal regions. Finally, Colacchio and Vergori (2022) analyze panel data of the Italian region from 1997 to 2019, proving the EDTG hypothesis. Furthermore, they quantify the impact of economic growth on the employment rate in the tourist sectors, which is in line with the employment intensity of growth for the whole economy.

From what has been said so far, it emerges that the empirical literature has also obtained mixed results about the causal link between growth and tourism for Italy. These

mixed results could depend on various factors such as the estimated period and its length, the methodologies, and the variables used as proxies. Furthermore, Italian data have usually been analyzed jointly with data about countries having common characteristics with Italy (e.g., Mediterranean countries and, more generally, European countries).

Our research proposes to take a forward step from the abovementioned literature by focusing exclusively on Italian data and using an economic variable as a proxy for domestic and inbound tourism. Through a PVAR model, we test two hypotheses that, in our opinion, deserve to be investigated. The first hypothesis concerns the role played by tourism in different Italian regions with varying economic development levels. In other terms, we believe that the nexus between tourism expansion and economic growth is strictly related to the specific characteristic of each area of the country. Corroborating this hypothesis shows the pathway for future research: it should be done at a subnational level if the aim is to fine tune policies to stimulate economic growth. Secondly, we hypothesize that the nexus between economic growth and tourism changes according to whether the period analyzed is a crisis period or not. Corroborating this hypothesis allows policymakers to understand if tourism has a pro-cyclical role.

### 3. Database Description

We use data collected by the Italian Institute of Statistics (ISTAT). Panel data concern the real total gross domestic product (PGDP) and the real tourism value added (PTVA) for the twenty Italian regions over the period 1997–2019[1]. Both variables are in per capita terms.

Following Ivanov and Webster (2013), we chose to use tourism value added as the proxy of tourism's contribution to the regional economies. We maintain that this choice is consistent with the variable we use to evaluate economic growth (GDP per capita), allowing us to consider, at the same time, the effects of both international and domestic tourism. Unlike most previous studies, we did not use tourism expenditure or receipts data as proxies of tourism development because they are available exclusively for international tourism, through a sample survey provided by the Bank of Italy. Although the relative importance of foreign tourists has grown over time, in 2019, domestic tourism in Italy was still about 50% of the total. Therefore, focusing exclusively on international tourism would imply neglecting the impact on the Italian economy of half of the tourism demand. Regarding tourist arrivals, we think that they are not a good proxy of the economic impact of tourism, since, for instance, an increase in the number of tourists does not automatically mean higher tourism expenditure at the destination.

Having said this, it is worth noting that PTVA is also not free from drawbacks. Its disadvantages stem from the difficulties in defining the economic sectors involved in the tourism experience: all consumption goods or services can potentially be part of tourism expenditure. Dealing with this thorny issue, many years of efforts by several national and international research institutions have led to the setting of standard guidelines to compile the Tourism Satellite Account (UN and UNWTO 2010), which provides recommendations for using a common reference framework in the compilation of tourism statistics. As a result, the main typical tourism sectors have been defined, and several countries have compiled their Tourism Satellite Accounts, albeit usually for only a few years and at the national level. Unfortunately, the need for regional statistics, stemming from the specific features of tourism across a country's regions, is still hampered by some statistical limitations in producing regional data (e.g., Frent and Frechtling 2020).

Despite the difficulties mentioned above, we estimated the relationship between tourism development and economic growth using regional data. The decision to perform our analysis at the NUTS 2 level lies in two main reasons. First, from an economic perspective, we could investigate the relationship between tourism development and economic growth for different macro areas. In turn, this allowed us to evaluate the "weight" of each geographical area in determining the relationship between the two phenomena for Italy as a whole. Second, from a methodological point of view, relying on panel data allowed us to obtain more accurate results than using longitudinal time series.



Our variable PTVA is the total value added of the "core tourism sectors" (Jones et al. 2003) of the Italian regions' economies: accommodation, food industry, arts, entertainment, and fun activities. We did not consider the transport sector for several reasons. First, at the regional level, data about transport value added are included in a broader category labeled "transport and storage," and it is not possible to disentangle transport from storage data. Second, even if we had that kind of data, they would have to be considered only to the extent their economic benefits affect the visited area under consideration. On the contrary, most transport expenditures are probably not ascribable to the regional economies (at the most, the expenses for national carriers benefit the national economy as a whole, but it is not easy to ascribe them to a specific region). It is worth noting here that the core of the tourism industry is accommodation and food and beverage services, which account for more than 75% of the Italian value added created in the tourism characteristic industries (according to Table 6 of the Italian Tourism Satellite Accounts for 2017).

On the contrary, the data of the three Tourism Satellite Accounts available for Italy show that transportation accounts for about 8% of the tourism value added and therefore, according to what we have just said, its relevance at the regional level should be lower than 8%. Accordingly, although not considering the transport industry could be a limit of our analysis, it should not compromise the results.

We are aware that the income produced by the sectors we have considered does not represent the entire income derived from tourism. However, for the reasons stated above, we maintain that using PTVA as a proxy of tourism's contribution to the regional economies stands as reasonable, considering the relevance of accommodation and food services upon the value added created by the core tourism sectors.

Table 1 shows some descriptive statistics about the PGDP and PTVA variables. Grouping the Italian regions according to the macro-area of membership (see the Appendix A) allows us to highlight the economic gap between the north and the south of the country. The gap is evident looking at both the PGDP and PTVA. Piedmont is the only region with a tourism value added similar to the southern regions, notwithstanding its higher GDP per capita (compared to the southern regions). Trentino Alto Adige and Valle D'Aosta have the two highest tourism value added.

**Table 1.** Descriptive statistics.

| | Regions | Variables | Mean | Stand. Dev. | Min | Max |
|---|---|---|---|---|---|---|
| **Northwest** | Liguria (LIG) | PGDP | 28,079 | 1494 | 25,291 | 30,559 |
| | | PTVA | 1618 | 102 | 1262 | 1758 |
| | Lombardy (LOM) | PGDP | 33,628 | 1225 | 31,108 | 35,667 |
| | | PTVA | 1223 | 59 | 1105 | 1318 |
| | Piedmont (PI) | PGDP | 27,185 | 1267 | 25,377 | 29,479 |
| | | PTVA | 974 | 76 | 844 | 1099 |
| | Valle Aosta (VA) | PGDP | 35,903 | 1882 | 32,224 | 38,335 |
| | | PTVA | 3341 | 310 | 2925 | 3954 |
| **Northeast** | Emilia Romagna (ER) | PGDP | 30,992 | 1304 | 28,287 | 33,407 |
| | | PTVA | 1387 | 76 | 1226 | 1556 |
| | Friuli V. G. (FVG) | PGDP | 27,079 | 1201 | 25,242 | 29,395 |
| | | PTVA | 1343 | 73 | 1241 | 1501 |
| | Trentino A.A. (TR) | PGDP | 35,441 | 904 | 33,334 | 36,876 |
| | | PTVA | 3731 | 417 | 3206 | 4471 |
| | Veneto (VNT) | PGDP | 28,526 | 1074 | 26,685 | 30,482 |
| | | PTVA | 1452 | 88 | 1317 | 1665 |
| **Center** | Lazio (LA) | PGDP | 31,362 | 2139 | 28,307 | 34,873 |
| | | PTVA | 1614 | 193 | 1109 | 1886 |
| | Marche (MA) | PGDP | 24,303 | 1275 | 22,130 | 26,779 |
| | | PTVA | 1099 | 62 | 948 | 1226 |

**Table 1.** *Cont.*

| | *Regions* | *Variables* | *Mean* | *Stand. Dev.* | *Min* | *Max* |
|---|---|---|---|---|---|---|
| | Tuscany (TO) | PGDP | 27,195 | 1106 | 24,975 | 29,097 |
| | | PTVA | 1494 | 100 | 1192 | 1607 |
| | Umbria (UM) | PGDP | 24,252 | 1801 | 21,086 | 26,409 |
| | | PTVA | 1123 | 81 | 1009 | 1286 |
| South and Islands | Abruzzo (A) | PGDP | 22,456 | 759 | 21,365 | 23,681 |
| | | PTVA | 1088 | 95 | 966 | 1376 |
| | Apulia (PU) | PGDP | 16,553 | 719 | 15,318 | 17,677 |
| | | PTVA | 697 | 85 | 531 | 854 |
| | Basilicata (B) | PGDP | 18,456 | 1111 | 15,905 | 20,255 |
| | | PTVA | 718 | 64 | 558 | 853 |
| | Calabria (CAL) | PGDP | 15,295 | 872 | 13,829 | 16,669 |
| | | PTVA | 727 | 55 | 612 | 841 |
| | Campania (CAM) | PGDP | 16,993 | 955 | 15,615 | 18,498 |
| | | PTVA | 873 | 62 | 787 | 983 |
| | Molise (MO) | PGDP | 19,480 | 1329 | 17,199 | 21,631 |
| | | PTVA | 853 | 77 | 721 | 972 |
| | Sardinia (SA) | PGDP | 18,594 | 841 | 16,979 | 19,877 |
| | | PTVA | 952 | 179 | 653 | 1270 |
| | Sicily (SI) | PGDP | 16,464 | 891 | 15,123 | 17,911 |
| | | PTVA | 785 | 97 | 532 | 905 |
| | ITALY | PGDP | 25,505 | 1098 | 23,499 | 27,428 |
| | | PTVA | 1199 | 65 | 1025 | 1276 |

Source: Own elaboration of ISTAT data.

## 4. A Preliminary Statistical Data Analysis

Since we aimed at investigating the relationship between tourism and the standard of living related to economic growth, the analysis focused on the growth rates of per capita real total gross domestic product (PGDPgr) and per capita real tourism added value (PTVAgr).

Before performing our analysis, some preliminary statistical tests needed to be completed. First, in dealing with panel data, it is essential to investigate the presence of cross-sectional dependence (CSD). The estimation of the "cross sectional exponent" with the xtcse2 STATA module (Ditzen 2019) states that there is evidence of strong CSD in our panel data, implying that the null hypothesis of weak CSD of the errors can be rejected according to the test by Pesaran (2015).

After that, we checked the integration order of our series using the so called "second generations tests", the Breitung test and the Pesaran cross-sectionally augmented Dickey–Fuller ADF statistics (CADF) (see Breitung and Das 2005; Pesaran 2003), which are robust to CSD. In Table 2, for the sake of brevity, we report only the main results of the Breitung test, which, however, are strongly consistent with those obtained by the CADF test.

**Table 2.** Breitung unit root test. $H_0$: panels contain unit roots.

| Variable | Lambda | *p*-Value | Integration Order |
|---|---|---|---|
| Ln(PGDP) | −0.5524 | 0.2903 | I(1) |
| ΔLn(PGDP) | −3.5690 | 0.0002 | I(0) |
| Ln(PTVA) | 0.0032 | 0.5013 | I(1) |
| ΔLn(PTVA) | −5.7369 | 0.0000 | I(0) |

As is evident from Table 2, the logs of PGDP and PTVA are integrated on order 1, while the log differences are stationary, being I(0). As a final step of this preliminary analysis, we have performed a cointegration test in order to investigate if there exists a long-run relationship between ln(PGDP) and ln(PTVA), relying on the "second-generation"

Westerlund cointegration test, which is robust against CSD. The results of this test, reported in Table 3, show that our variables are not cointegrate[2].

**Table 3.** Westerlund cointegration test.

| Statistic | Variables: ln(PGDP) and ln(PTVA)-$H_0$: No Cointegration | | |
| | Value | Z-Value | *p*-Value |
| --- | --- | --- | --- |
| Gt | −0.648 | 1.411 | 0.921 |
| Ga | −0.971 | 2.784 | 0.997 |
| Pt | −1.028 | 1.075 | 0.859 |
| Pa | −0.260 | 1.184 | 0.882 |

*Granger Causality Test*

To test for the Granger causality, we have used the so-called second-generation tests, robust to CSD, relying on the STATA package xtgranger (see Xiao et al. 2021). In particular, we performed the Half-Panel Jackknife (HPJ) Wald-type test, also taking into account a bootstrap variance estimator. We obtained consistent results from these tests, according to which there is Granger causality only from PGDPgr to PTVAgr, but no Granger causality was detected in the opposite direction.

**5. The Model**

Since we have been dealing with data concerning various regions of the same country for two decades, we chose to perform an estimation in the form of a panel VAR (PVAR) model, which is a combination a dynamic panel model and a vector autoregressive model, more efficient than aggregate time series analysis (Hsiao 2007).

Before testing for the stationarity of both time series, we have selected the optimal lag length according to the AIC(n), SC(n), HQ(n), and FPE(n) criteria. In accordance with HQ and SC criteria the optimal lag length is 1, while AIC and PFE criteria suggest that the optimal length is 4. We chose the optimal length equal to 1 because it is the usual choice with annual data and because the number of time periods of our panel is relatively small.

Since several tests that we have performed suggest choosing a "two-ways" effects model, including both time and cross-section dummies, the PVAR to be estimated is the following one:

$$PTVAgr_{it} = \alpha_{11}PTVAgr_{it-1} + \alpha_{12}PGDPgr_{it-1} + \lambda_t + \mu_i + \varepsilon_{1,t}$$
$$PGDPgr_{it} = \alpha_{21}PTVAgr_{it-1} + \alpha_{22}PGDPgr_{it-1} + \lambda_t + \mu_i + \varepsilon_{2,t}$$

(1)

where $\lambda_t$ and $\mu_i$ are respectively the time and the cross-section dummies.

The structure of our panel from one side—where both T and N are "sufficiently large"—and the above mentioned presence of strong CSD from the other side led us to adopt the following strategy: we first estimated each equation of model (1) employing an estimator—suited for dynamic panel data—robust to CSD, after which we used a LSDV PVAR estimator in order to obtain the impulse response functions (IRF). In particular, we relied on a bootstrap corrected fixed effects (BCFE) estimator proposed by De Vos et al. (2015) and, for robustness check, we also performed an alternative estimation using a panel-corrected standard error estimator (PCSE)[3]. The results of our estimations—including the LSDV estimations—are reported in the following table, Table 4.

In line with the Granger causality analysis performed before, the above regression results confirm a significant dependence of $PTVAgr_t$ on $PGDPgr_{t-1}$. Furthermore, an autoregressive structure of PTVAgr, with a significant negative coefficient, emerges.

**Table 4.** Model estimation.

| | **PTVAgr$_t$** | | | **PGDPgr$_t$** | | |
|---|---|---|---|---|---|---|
| | *BCFE* [1] | *PCSE* | *LSDV* | *BCFE* [1] | *PCSE* | *LSDV* |
| PTVAgr$_{t-1}$ | −0.1783 ** | −0.1605 * | 0.2155 *** | −0.0179 | −0.0194 | −0.0174 |
| | (0.0866) | (0.0840) | (0.0502) | (0.0199) | (0.0179) | (0.0165) |
| PGDPgr$_{t-1}$ | 0.2749 * | 0.2887 * | 0.2829 * | 0.0012 | −0.0390 | −0.0403 |
| | (0.1498) | (0.1494) | (0.1516) | (0.0439) | (0.0858) | (0.0514) |

Notes: [1] Bootstrapped standard errors; inference performed with non-parametric bootstrap. *** = *p*-value < 1%; ** = *p*-value < 5%; * = *p*-value < 10%.

### 5.1. Impulse Response Functions Analysis

After having checked the stability of the model (the eigenvalues are negative and lie inside the unit circle), we analyzed the generalized impulse response functions.

From Figure 1 (see the upper-right box), we can see the response function of PTVAgr to a one-time (one standard deviation) positive shock on PGDPgr. After an initial significant increase in the PTVAgr value, the response function curves downward, and the shock is "reabsorbed" very slowly in about four years. On the contrary, a one-time shock on PTVAgr (see the lower-left box) causes an initial slight decrease in PGDPgr that is reabsorbed in about four years. These findings are in line with the analysis of the previous section and confirm the relevance of the impact of GDP changes on value-added per capita tourism[4]. In addition, the oscillating behavior of the response functions is in line with the negative sign of the estimated eigenvalues.

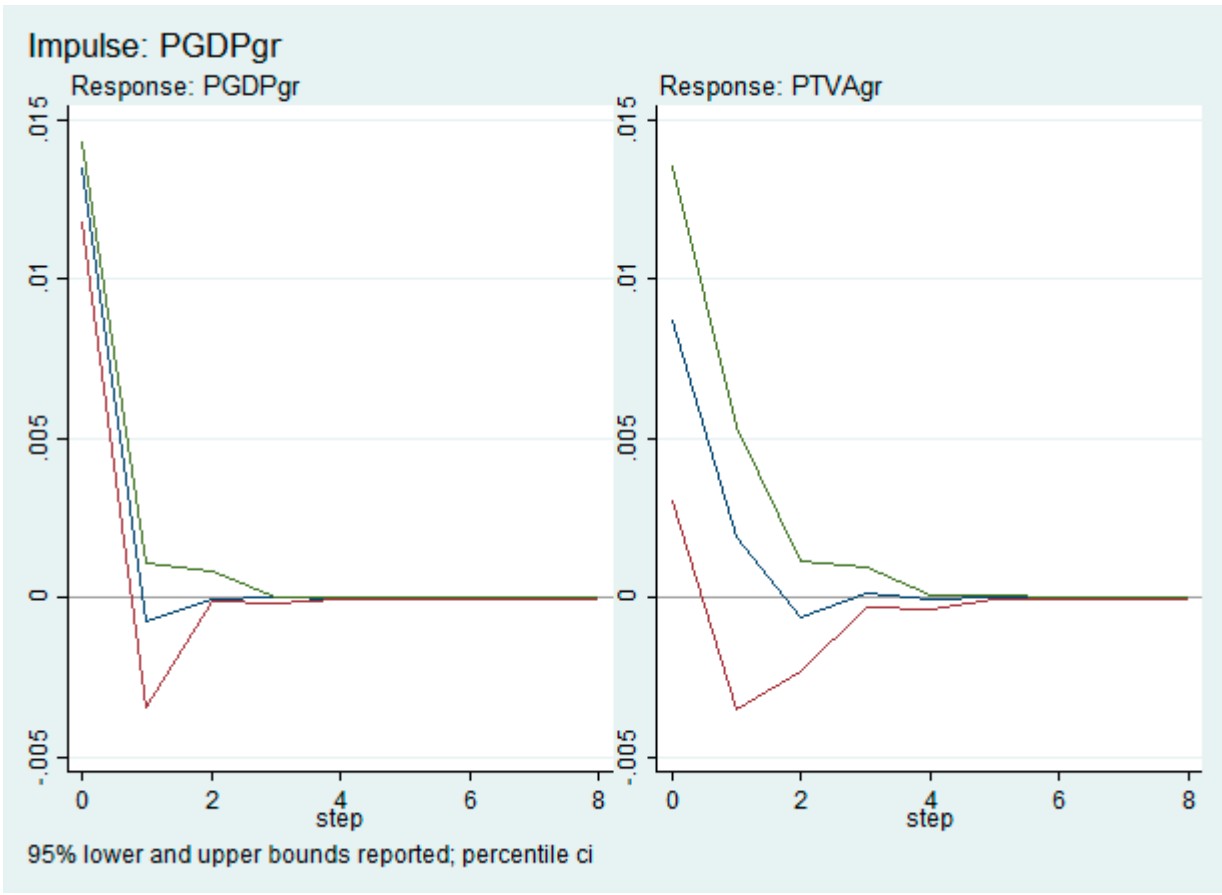

**Figure 1.** *Cont.*

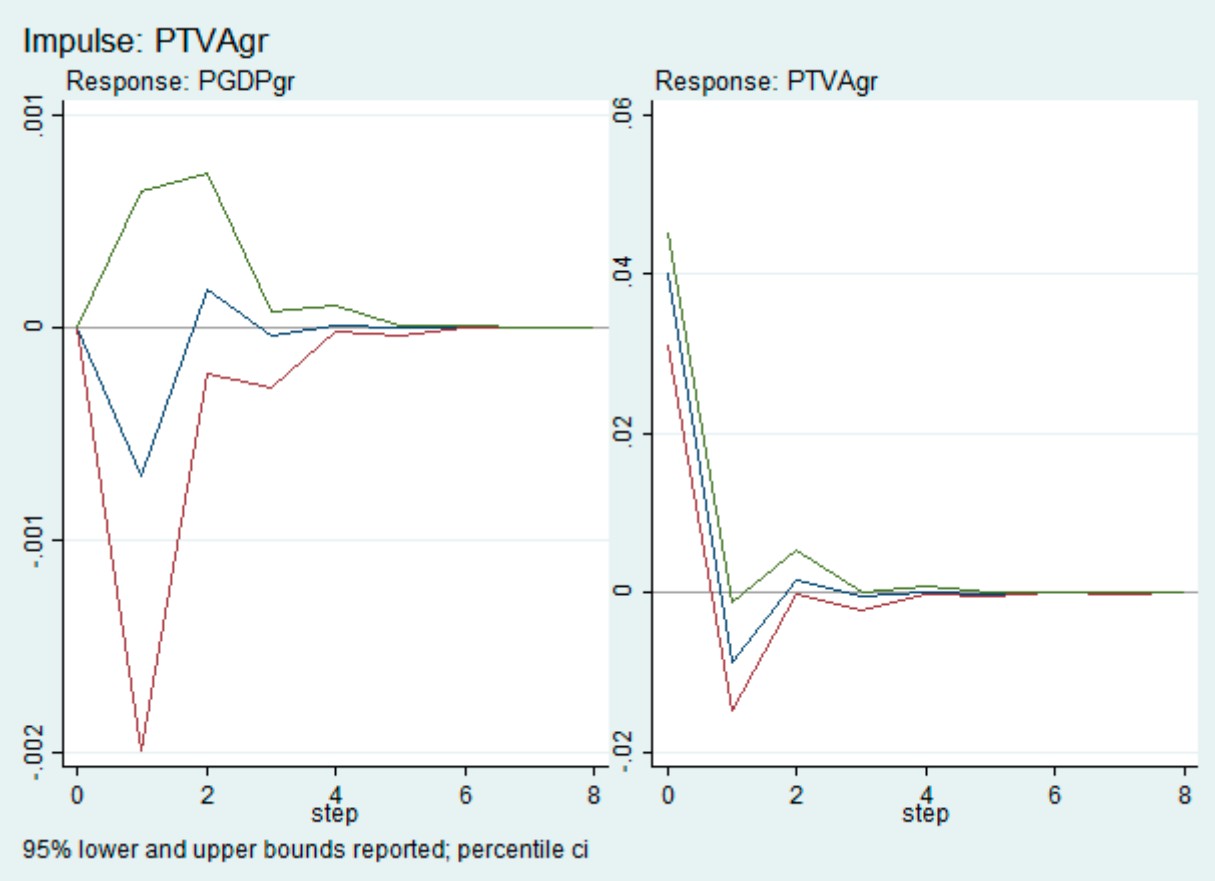

**Figure 1.** Impulse response functions.

Before concluding this section, we point out to the reader that in order to provide further robustness checks, we have estimated alternative models, replacing for instance the PGDPgr with the GDP growth rate *tout court*, or replacing PTVAgr with the growth rate of tourist overnight stays, etc. The main results of these alternative analyses—which can be provided upon request—are almost identical to the outcomes showed in Table 3, confirming once again the EDTG hypothesis for the period under examination.

### 5.2. Subperiod Analysis

To perform a more in-depth analysis of the relationship between economic growth and tourism development, we split the original sample into two equal subperiods, 1998–2008 and 2009–2019. We maintain that this investigation should allow us to understand how the results we found in the previous section are related to some economic conditions peculiar to each subperiod. The results shown in Table 5a,b, where for the sake of brevity we report only the BCFE and PCSE estimates, demonstrate that the dependence of tourism development on economic growth has its roots in the period 2009–2019. It is worth pointing out that this subperiod has been characterized by two exogenous shocks that severely affected the Italian GDP growth rate: the "subprime" economic crisis (2009) and the economic recession engendered by the so-called sovereign debt crisis (2012).

The autoregressive structure of PTVAgr is confirmed during both subperiods, even if it is unambiguously statistically significant only in the second subperiod. It is worth noting that during 2009–2019, an autoregressive structure for real per capita GDP growth rate emerges too.

**Table 5.** (**a**): Estimated models for Italy, years: 1998–2008; (**b**): Estimated models for Italy, years: 2009–2019.

| | a: Estimated Models for Italy, Years: 1998–2008 | | | |
|---|---|---|---|---|
| | **PTVAgr$_t$** | | **PGDPgr$_t$** | |
| | BCFE [1] | PCSE | BCFE [1] | PCSE |
| PTVAgr$_{t-1}$ | −0.1994 ** | −0.1519 | −0.0311 | −0.0318 * |
| | (0.0957) | (0.1384) | (0.0317) | (0.0192) |
| PGDPgr$_{t-1}$ | 0.1080 | 0.1515 | 0.1291 | 0.0890 |
| | (0.3480) | (0.2707) | (0.0840) | (0.1393) |
| | b: Estimated Models for Italy, Years: 2009–2019 | | | |
| | **PTVAgr$_t$** | | **PGDPgr$_t$** | |
| | BCFE [1] | PCSE | BCFE [1] | PCSE |
| PTVAgr$_{t-1}$ | −0.2146 ** | −0.3047 *** | −0.0141 | −0.0200 |
| | (0.0876) | (0.1088) | (0.0403) | (0.0271) |
| PGDPgr$_{t-1}$ | 0.2709 ** | 0.3549 *** | −0.1161 ** | −0.1356 ** |
| | (0.1226) | (0.0845) | (0.0538) | (0.0632) |

Notes: [1] Bootstrapped standard errors; inference performed with non-parametric bootstrap. *** = $p$-value < 1%; ** = $p$-value < 5%; * = $p$-value < 10%.

### 5.3. Cluster Analysis

To explore in more detail the unidirectional relationship from economic growth to tourism development emerging in the previous analysis, we clustered the Italian regions according to their level of economic development. We did not define clusters in advance, but we derived them following the K-means clustering partition method—based on the two-dimensional classifications of the level of real per capita GDP and the level of real per capita tourism added value (PTVA). After having standardized the values of both PGDP and PTVA, we estimated the optimal number of clusters (equal to 4), and then we applied the K-means algorithm (see Kassambara 2017): the results of our analysis are summed up in the following figure, Figure 2[5].

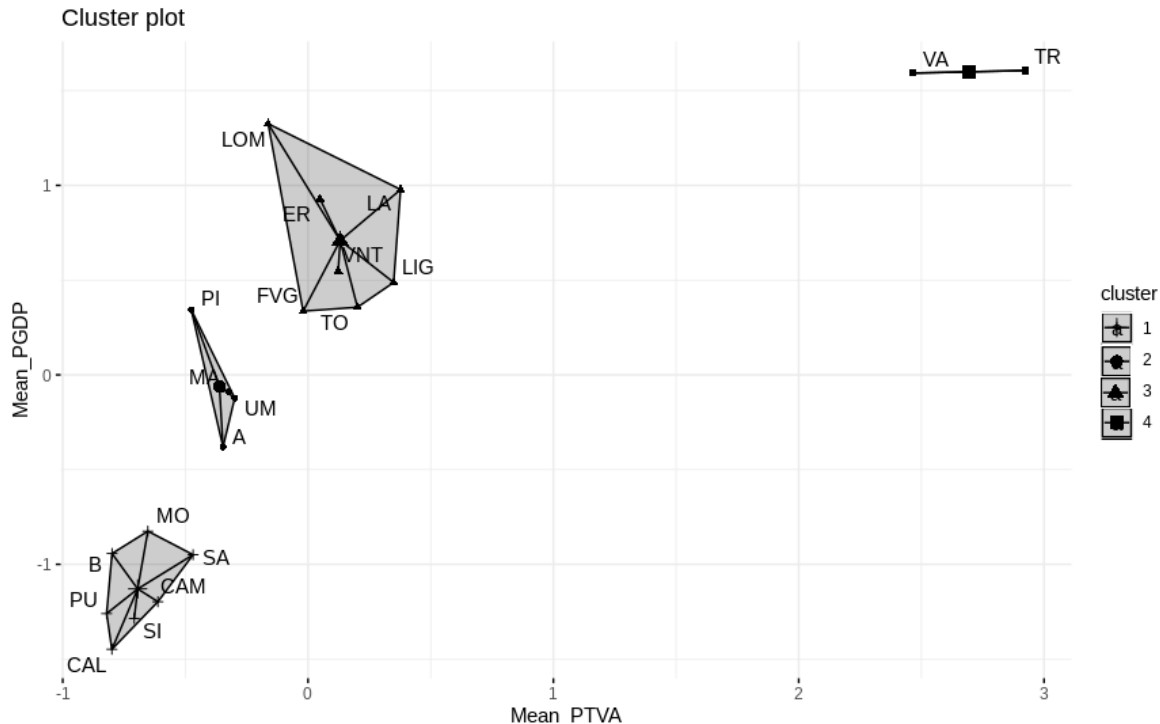

**Figure 2.** Cluster analysis.

From Figure 2 emerges that cluster 1 (CL1 from now on) involves all the southern Italian regions apart from Abruzzo. Abruzzo is usually merged into south Italy—as officially recognized by Italian and European Institutes of statistics—even if it is geographically considered part of central Italy (see the map in the Appendix A). Furthermore, Abruzzo's GDP levels are closer to those of the central Italian regions than to those of the southern regions. The central and northern Italian regions are distributed among the other three clusters. In particular, the wealthiest Italian regions belong to cluster 3 (CL3 hereafter). Cluster 4 (CL4 hereafter) involves the wealthiest Italian regions with a Special Statute, while cluster 2 (CL2 hereafter) is made up of four regions, two of which are central regions.

In light of the heterogeneity that characterizes the various areas of the country, we carried out a deeper investigation of the four clusters described in the previous section, estimating, first of all, the model (1) for the CL1, CL2, and CL3 clusters. Given the structure of the 3 panels, we replaced the PCSE estimator with a feasible generalized least square estimator (FGLS), robust to cross-sectional correlation (the so-called Parks method), which is more accurate and efficient than the PCSE estimator when the T/N ratio is sufficiently large (as in our case)[6]. Table 6a–c sum up the main results of our analysis.

**Table 6.** (**a**): Estimated models for cluster CL1; (**b**): Estimated models for cluster CL2; (**c**) Estimated models for cluster CL3.

| | a: Estimated Models for Cluster CL1 | | | |
|---|---|---|---|---|
| | $PTVAgr_t$ | | $PGDPgr_t$ | |
| | *BCFE* [1] | *FGLS* | *BCFE* [1] | *FGLS* |
| $PTVAgr_{t-1}$ | −0.2260 * | −0.2159 ** | −0.0143 | −0.0089 |
| | (0.1266) | (0.0878) | (0.0420) | (0.0185) |
| $PGDPgr_{t-1}$ | −0.1862 | 0.1427 | 0.0908 | 0.0372 |
| | (0.2133) | (0.2053) | (0.0740) | (0.0904) |
| | b: Estimated Models for Cluster CL2 | | | |
| | $PTVAgr_t$ | | $PGDPgr_t$ | |
| | *BCFE* [1] | *FGLS* | *BCFE* [1] | *FGLS* |
| $PTVAgr_{t-1}$ | −0.0333 | −0.0358 | −0.0414 | −0.0492 |
| | (0.1291) | (0.1653) | (0.0325) | (0.0310) |
| $PGDPgr_{t-1}$ | 0.1042 | 0.6920 | −0.0482 | 0.0413 |
| | (0.4599) | (0.6384) | (0.2813) | (0.1266) |
| | c: Estimated Models for Cluster CL3 | | | |
| | $PTVAgr_t$ | | $PGDPgr_t$ | |
| | *BCFE* [1] | *FGLS* | *BCFE* [1] | *FGLS* |
| $PTVAgr_{t-1}$ | −0.1516 | −0.2079 ** | 0.0022 | 0.0041 |
| | (0.0984) | (0.0988) | (0.0375) | (0.0297) |
| $PGDPgr_{t-1}$ | 0.4063 ** | 0.5400 ** | 0.0478 | −0.0295 |
| | (0.1908) | (0.2568) | (0.0827) | (0.0975) |

Notes: [1] Bootstrapped standard errors; inference performed with non-parametric bootstrap. *** = *p*-value < 1%; ** = *p*-value < 5%; * = *p*-value < 10%.

Our results highlight a statistically significant dependence of the per capita tourism value-added growth rate on the lagged real PGDP growth rate only for cluster CL3. The positive signs indicate a positive lagged effect of economic growth on tourism, and the magnitude of this effect is quite large.

The negative autoregressive structure of per capita tourism value-added growth rates is corroborated for all three clusters, but it is statistically significant only for CL1.

As to the fourth small cluster, performing a PVAR analysis with only two regions makes no sense. Accordingly, we have simply carried out two VAR estimations, one for each region. In both cases, the parameters of lagged values $PTVAgr_{t-1}$ and $PGDPgr_{t-1}$ are

not statistically significant, which means that there is no effect of $PTVAgr_{t-1}$ on $PGDPgr_t$ and of $PGDPgr_{t-1}$ on $PTVAgr_t$.

## 6. Discussion

Our results highlight a clear dependence of tourism development on economic growth.

A first reasonable explanation of our findings may be related to the fact that economic expansion could improve the tourism-related infrastructures and service quality, attracting new domestic and international demand. A second explanation could be based on a dynamic version of the Keynesian theory of consumption demand. Since we are dealing here with both domestic and foreign demand for tourism services, it is reasonable that an increase in the past value of GDP causes an increase in current domestic consumption and hence in current domestic demand for tourism services. However, in a country where foreign tourism is 50% of the total, one would expect to find, at the same time, a "multiplier" effect going in the opposite direction, e.g., from $PTVAgr_{t-1}$ to $PGDPgr_t$, since tourism demand from foreigners can be considered as an autonomous component of national aggregate demand. This expectation has been disregarded, although the value of foreign tourism expenditure in Italy has been growing in the last decade (www.bancaditalia.it, accessed on 27 February 2023).

When we investigate the relationship between tourism development and economic growth in the subperiods 1998–2008 and 2009–2019, we find that the dependence of tourism on economic growth has its roots exclusively in the second subperiod. This result seems to be in line with Shahzad et al. (2017), who found that the link between tourism and economic growth for Italy is stronger during the recession periods. It may be interesting to recall to the reader that the intense contractionary economic phase of 2011–2013 was accompanied by a decrease in domestic tourism demand, with a consequent reduction in tourism value added, while, during the economic recovery that took place from 2014 onwards, tourism demand increased at higher growth rates compared to the first subperiod. Furthermore, foreign tourism overnight stays have grown more from 2009 to 2019 than from 1998 to 2008, surpassing the domestic overnight stays in Italy for the first time in 2017.

The analysis at the subnational level, through the definition of regional clusters, allows us to identify the CL3 cluster as the one that mainly contributes to the validation of the EDTG hypothesis for the whole country. It is worth recalling that this cluster brings together the wealthiest Italian regions where the role played by the manufacturing is higher than the Italian average and where most of the service sector's value-added concerns insurance, financial, real estate, and administrative services (see: www.dati.istat.it, accessed on 13 January 2023). Second, although six out of seven regions belonging to CL3 are among the top ten Italian tourist destinations, all of them are among the top ten Italian regions for foreign visitors travelling for business purposes (Bank of Italy 2019). Business tourism, indeed, tends to be related to the GDP level of the destination country other than the GDP of the tourists' country of origin (Kulendran and Witt 2003). Accordingly, although CL3 attracts the majority of tourist flows in Italy and about 72% of foreign tourist expenditure, tourism is not the most important economic activity for the area; thus, in relative terms, it is probably not so relevant to "drive" economic growth.

The results obtained for CL1 and CL2 clusters exclude the existence of a relationship between tourism development and economic growth. It sounds not so strange for the CL2 cluster if one thinks that it registers less than 10% of the tourist overnight stays in Italy, and, in this respect, its relative importance has decreased in the period analyzed. Furthermore, according to the bank of Italy's data, it attracts about 6% of the foreign tourist expenditures in Italy. On the contrary, about the CL1 cluster, it is worth noting that the increase in tourist overnight stays by more than 50% is mainly due to foreign tourism. The share of value added in the tourist sector (according to our definition) on the total GDP increased by one percentage point in 2019 compared to 1998 (against 0.3% for CL3 and 0.5% for CL2). Although southern regions (that, as we have already hinted, overlap with the CL1 cluster) attract a small share of foreign tourist expenditures in Italy, its relative importance in the

Italian scene increased from 8.3% in 1998 to 14.2% in 2019 due to an increment in foreign tourist expenditures of more than 180% in the period considered (against 55% for CL3 and 49% for CL1). However, despite the growth in the tourism phenomenon, in the south of Italy, no empirical relationship with economic growth appears to be statistically significant yet. In this respect, some considerations should be made. First, foreign tourism increased more than domestic tourism in 2019 compared to 1998. Since it is mainly leisure tourism, it depends, among others, on the GDP of the countries of origin. Furthermore, according to a recent study (Vergori and Arima 2022), the advent of low-cost carriers has contributed to the increase in foreign tourism in some peripheral (southern) Italian regions more than in other parts of Italy. Having said that and considering also that the economic growth of the CL1 cluster has been slower than the CL3 one, it is reasonable that regional GDP could play a marginal role in fostering foreign tourism. On the other hand, notwithstanding its development, tourism is not a leading sector for CL1 cluster.

Coming to the CL4 cluster, the results obtained in the previous section support the "neutrality hypothesis" for both Valle D'Aosta and Trentino Alto Adige. In an attempt to explain these outcomes, one has to consider that both regions, with particular regard to the tourism sector, are strongly integrated with their respective bordering transalpine areas (see Bank of Italy 2019, pages 48 and *ff*.). We think that this fact may help to explain the negligible effect of the domestic GDP growth rates on the expansion of tourism value added in these regions. In addition, over the period 2000–2018, for both regions, the value of PTVA has been roughly constant, with an average PTVA/PGDP ratio that is almost double the average Italian value, suggesting that the tourism sector potentiality may have already been, at least in part, exploited.

## 7. Conclusions

The analysis we have performed in the previous sections suggests that economic growth boosts the Italian tourism industry, though tourism is not a leading sector for Italy.

Apart from considering both domestic and international tourism, another crucial aspect of our study lies in the investigation of the relationship between tourism development and economic growth at the subnational level. The interesting aspect of our cluster analysis is that it shows how different areas of the country contribute to confirming the final hypothesis for Italy as a whole. Our results suggest that the link between tourism expansion and economic growth is strictly related to the specific character of each area. The heterogeneity of the Italian economy (and more generally, of the economy of every nation) necessarily calls for the fine tuning of proper policies that can meet the needs of each region.

According to our results, policymakers cannot rely heavily on the tourism sector to foster economic development. For the CL3 cluster, the increase in the GDP growth rate has constituted the real source for further development of the tourism industry. The reason why tourism is not a leading sector for those economies probably lies in the sector's lower importance compared to other more productive sectors. This should not be a problem considering that they are the wealthiest Italian regions.

The southern regions (CL1) are peripheral with respect to northern Europe and are the poorest, but they are very attractive and have a mild climate. The absence of any relationship between economic growth and tourism development in the period analyzed does not mean there is no potential to create a tourism sector that drives economic growth. All the regions belonging to the CL1 cluster—except for Molise, which, however, is one of the smallest regions in Italy—were characterized by a continuous increase in the growth rate of per capita tourism value added, which has led to an increase in the tourism value-added share (of GDP). However, at the same time, this change in the composition of GDP has had no significant effect on the economic performances of southern regions. We maintain that southern Italian regions need a more efficient and stronger economic environment, comparable to northern European standards, before tourism can become a leading sector. It is evident that the Italian government should, first and foremost, improve the region's

infrastructure, which is essential for access to a territory, especially if it is peripheral, promoting at the same time enterprise innovations and human capital accumulation, since the increase in the supply of a competent, professional, skilled workforce is another crucial precondition for fostering the growth of the tourism sector. Clearly, the political parties representing employers should be actively involved in this type of process, in an attempt to channel private investments towards a tourism provision with higher value added.

These findings give clear pointers for policy on future resource allocations since they indicate that one cannot simply think of boosting economic growth through the increase in the tourism sector.

Finally, the economic divide between north and south also characterizes other European countries. As pointed out by De Grauwe (2020), as market integration between countries proceeds, national borders become less and less important as a factor that decides the location of economic activities; hence, it is crucial not to neglect the most peripheral and "poorest" regions that could pay the consequences of integration with the richest areas. It is worth mentioning in this regard that the "Next Generation EU program"—the recovery plan agreed to by the European Council to support European countries hit by the Covid-19 pandemic—allocated about 205 billion euros to Italy to be spent over the period 2021–2026. The Italian government envisages allocating 6.68 billion euros to culture and tourism, considered among the key sectors in enabling the Italian economy to recover from the crisis generated by the pandemic. However, in line with what we have argued so far, tourism is not a leading sector for the Italian regions. We therefore maintain that the European funds should mainly be used to reduce the economic gap between the north and the south of Italy. Once proper investments are made, in fact, tourism can potentially lead to the growth of southern economies.

To conclude, the main results of our investigation show that for the country as a whole, tourism growth depends on the past value of the economic growth rate, especially for the subperiod 2009–2019, while tourism is not a leading sector for Italy. In addition, our cluster analysis clarifies that this outcome—i.e., the validation of the EDTG hypothesis—is strongly determined by the cluster that covers the wealthiest Italian regions.

A final consideration about the limitations of our study inevitably comes up against data availability issues. The tourism value-added variable we used to measure tourism development does not consider the value added created by the transport sector because data at the NUTS 2 level are not available. Although, as we have already explained in the text, transportation should account for a small part of the regional tourism value added, the availability of those data would have allowed our results to be even more accurate.

**Author Contributions:** Both authors contributed to the research design, the analysis of the results, and the writing of the manuscript. G.C. has carried out the model implementation. All authors have read and agreed to the published version of the manuscript.

**Funding:** This research received no external funding.

**Conflicts of Interest:** The authors declare no conflict of interest.

**Appendix A  Map of Italy**

## Notes

[1] We point out to the reader that at the time we were writing this paper, data for the years 2021–2022 were not fully available. As for 2020, we decided not to consider this year because of the structural break caused by the COVID-19 pandemic.

[2] We point out to the reader that for the robustness check, we have also performed the alternative Pedroni cointegration test, whose results are, however, strongly consistent with those reported in Table 3.

[3] The Stata packages employed are, respectively, xtbcfe, xtpcse and xtvar.

[4] As is well-known, the presence of lagged dependent variables causes the OLS estimator to be biased—the so-called "Nickel bias"—which is of an order of $1/T$, where $T$ is the number of observations, i.e., the number of years in our case. For this reason, for the robustness check, we have also estimated model (1) throughout a Bayesian PVAR, which should alleviate this bias (see Weale and Wieladek 2014): we point out to the reader that in this case as well, the estimated results (and the IRF) are almost identical to those reported in Table 4 (and in Figure 1).

[5] For the robustness check, we have also used a different clustering method, performing a principal component analysis (PCA) on our dataset and then constructing a hierarchical clustering on principal components (HCPC). However, the results of this alternative analysis are consistent with those obtained through the K-means method.

[6] We used the xtgls Stata package. On the efficiency and accuracy of PSCE and FGLS estimators, the reader is referred to the pioneer contribution by Beck and Katz (1995), where the authors prove—through a Monte Carlo analysis—that the accuracy and efficiency of FGLS increase as the T/N ratio increases, to the point that this estimator should be preferred to PSCE when $T/N \geq 3$.

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
