# Peer review of "Tourism Development and Italian Economic Growth: The Weight of the Regional Economies"

_jrfm, doi:10.3390/jrfm16040245_

Round 1

Reviewer 1 Report

The authors aim to empirically investigate the relationship between tourism and economic growth in twenty Italian regions from 1997 to 2019. The novelty of their model lies in the use of tourism value added as a variable for tourism. This approach has the advantage of considering not only international tourism, which is the focus of the majority of papers in the field, but also domestic tourism. In my view, this is a more appropriate approach for developed countries compared to studies that only consider international tourism. Furthermore, the model's structure is clear and consistent. In the technical part of the paper, the authors provide sufficient information and explanations to justify their empirical methodology. However, the English in the paper could be slightly improved, as it is sometimes awkward.

One issue is that the authors sometimes use the abbreviation "PTVA," which makes sense, but at other times use "PTAV," which is obviously incorrect. However, this is a minor mistake that can be easily corrected.

Overall, I believe the paper should be accepted for publication once these minor mistakes are corrected.

Author Response

Many thanks for your comments. We replaced PTAV with PTVA in the text, we did not realize the typo. We asked a native language to check the English.

Reviewer 2 Report

1. Please add research question in start of the introduction, so that reader can understand your main concern and interest.

2. Please identify the research problem in better manner.

3. Please provide contribution in the last part of the introduction.

4. Implications are not provided in the introduction and in conclusion part fo the paper.

5. The is missing the hypotheses development, I suggest to add in literature review part.

6. The results presentation is not good, that must should be improved.

Author Response

Many thanks for your suggestions. According to your comments, we made many changes to the sections: Introduction, Literature review, and Conclusions. We hope these changes improved our discussion in line with your comments and suggestions. Furthermore, we asked a native language to check the English.

Reviewer 3 Report

Thank you for allowing me to review this manuscript. This study provides new insights into the relationship between economic growth and tourism development, using the value-added in the main economic sectors involved in tourism activity as a proxy for tourism development. The use of tourist value-added to analyze the effect of international and domestic tourism on per capita GDP growth is a novel approach to assessing the impact of the tourism sector on the overall economy. I am impressed by the thoroughness of this study, and I believe the findings offer significant value to tourism and economic development. However, there are some crucial questions and concerns that the author(s) should consider:

The abstract 

Abstract lacks results; please add two sentences about your panel analysis results. 

The introduction: 

This sentence, "The objective difficulties in measuring the economic impact of tourism have not prevented scholars from studying the effects of tourism growth on economic development." Look vague without reflection on any paragraphs; please link it to any paragraphs in your argument. 

You mentioned that "TSA assesses only the direct economic impact, defined as the value-added," You highlighted your contribution in the abstract that you would study the added value. So, I suggest avoiding the confusion by elaborating more on what TSA did and what you will contribute. Do you contribute to exploring the indirect effects considering the added value? Please elaborate on that part. 

Also, I suggest you add two more sentences on why you used GDP as a proxy for your panel data. 

The literature review: 

here, there is a vital concern that you should address; you write what previous literature did without synthesizing it or providing any hypotheses; what are your hypotheses? Your literature review ends abruptly without any conclusions and articulation, setting a straightforward research question. Your review could at least lead to the development/formulation of two /three open-ended research questions that could guide the data collection and analysis process.

The data elaboration and the results are well-designed; I have no problem with them. 

The discussion and conclusion parts 

These parts do not indicate how this differs from previous literature or suggest any theoretical implications for this research approach. It offers little discussion on the managerial implications of the findings. While the study identifies the role played by different areas of the country in determining the final hypothesis proved for Italy as a whole, there is no clarity on how this should influence managers' decisions or policies. Therefore, I suggest you highlight the theoretical contribution and managerial implications by reflecting on it beside its parts.

Author Response

Many thanks for your comments. We added the main results in the abstract. The sentence you considered vague is connected to the previous and the subsequent sentences, we tried to highlight it in the text.

About TSA, you are completely right. We preferred to delete the sentence about the direct impact of tourism because it causes confusion. We mentioned TSA because it is the most accurate estimation of tourism's impact on an economy, but it is performed only at the national level and just for a few years.

We have modified, according to your suggestions, the abstract, the end of the section dedicated to the literature review, the discussion, and the concluding sections.

Reviewer 4 Report

Thank you for allowing me to review this manuscript. I find the current manuscript exciting and enjoy reading it. However, I have a few suggestions to improve the manuscript:

Abstract

The implications and limitations of the study should be summarized in a few sentences and included in the abstract.

Introduction

Several sentences (e.g., first three sentences) need in-text references or citations from the first sentence until the end of the manuscript. We should be more careful in-text cite or references, especially when we are emphasizing or showing the importance of something.

The last paragraph needed to be more straightforward and understandable. Please, clearly mention how you fill those gaps and emphasize the purposes. How can this study contribute to the tourism field? Briefly, I suggest the authors on a captivating introduction, including the research problem and study contribution.

Literature Review

No concern in there.

Methods:

Why does the data captivate only the period of 1997-2019? We are in 2023; why did the authors not include the following years: 2020,2021,2022?

Results:

No concern in there.

Conclusion:

As a final suggestion, the limitations and theoretical implications of the study should be included (almost absent). The authors only mentioned the practical consequences, but they should also mention theoretical implications and limitations.

I wish the author the best of luck with the revision.

Author Response

Thank you for your comments. Following your suggestions, we modified the abstract, the introduction, and the conclusions. We added some references in the text (the sentences without references are personal considerations or generally recognized assumptions). We explained in a note why we analyzed data until 2019: data are available until 2020, but we chose not to consider the last year because of the shock due to the pandemic. We believe that considering the effects of the pandemic requires data at least for 2021 and 2022 too. Furthermore, we asked a native language to check the English.

Round 2

Reviewer 2 Report

Authors have incorporated the suggestions in good manners.

Author Response

Many thanks.